# High Accuracy Buoyancy for Underwater Gliders: The Uncertainty in the Depth Control [note 1]

**DOI:** 10.3390/s19081831

**Published:** 2019-04-17

**Authors:** Enrico Petritoli, Fabio Leccese, Marco Cagnetti

**Affiliations:** Science Department, Università degli Studi “Roma Tre”, Via della Vasca Navale n. 84, 00146 Rome, Italy; e_petritoli@libero.it (E.P.); ing.marco.cagnetti@gmail.com (M.C.)

**Keywords:** uncertainty, buoyancy, depth control, accuracy, AUV, glider, autonomous, underwater, vehicle

## Abstract

This paper is a section of several preliminary studies of the Underwater Drones Group of the Università degli Studi “Roma Tre” Science Department: We describe the study philosophy, the theoretical technological considerations for sizing and the development of a technological demonstrator of a high accuracy buoyancy and depth control. We develop the main requirements and the boundary conditions that design the buoyancy system and develop the mathematical conditions that define the main parameters.

## 1. Introduction

This paper is part of several preliminary studies by Underwater Drones Group (UDG) of the Science Department of the Università degli Studi “Roma Tre”, which is developing an advanced Autonomous Underwater Vehicle (AUV) for the exploration of the sea at high depths. The final aim of the project is to create a platform for underwater scientific research that can accommodate a wide range of different payloads. 

We will examine the buoyancy system and evaluate its sizing; then we will illustrate the technological solution we have come up with in order to realize the hydraulic system to be assembled in the Underwater Glider Mk. III (see Figure 1) [1,2,3,4,5].

### 1.1. The Underwater Glider

#### 1.1.1. AUV Evolution

The exploration of the underwater world has always been one of mankind’s dreams: submarines and bathyscaphes (for extreme depths) have been developed to study the “deep blue”. Due to obvious dangers, the human exploration can take place only for very short periods and very limited areas: for these reasons, the exploration of the sea has immediately been drawn towards unmanned automatic systems [6,7,8]. 

An AUV is a vehicle that travels underwater without requiring input from an operator; this means that it must be equipped with a “brain” that regulates and coordinates its position, its depth and its speed: moreover, it is able to collect and store data from the payload. One of the first realizations was the Autonomous LAgrangian Circulation Explorer (ALACE) system, a buoy that was able to vary its buoyancy and therefore its depth. Although it possessed a great endurance, it only could be employed for great depths and in open sea—the consequences of these limitations are evident. 

The next step was the use of Remote Operated Vehicles (ROVs). These, thanks to the constant development of electronic miniaturization, are extremely high performing vehicles for short-lasting marine operations, but the require the constant presence of a support vessel. 

The need to get rid of the randomness of the currents has led to the natural development of the underwater glider concept [9,10,11,12,13,14,15].

#### 1.1.2. The Underwater Glider

An underwater glider is a vehicle that, by changing its buoyancy, moves up and down in the ocean like a profiling float [16]. It uses hydrodynamic wings to convert vertical motion into horizontal motion, moving forward with very low power consumption [17,18,19,20,21,22]. While not as fast as conventional AUVs, the glider, using buoyancy-based propulsion, offers increased range and endurance compared to motor-driven vehicles and missions may extend to months and to several thousands of kilometres in range. An underwater glider follows an up-and-down, sawtooth-like mission profile providing data on temporal and spatial scales unavailable with previous types of AUVs [23,24,25,26,27].

#### 1.1.3. The Mk. III Architecture

The Mk. III sub-glider has a cylindrical fuselage with a radome on the bow containing the customizable payload and, on the other end, the hydrodynamic fairing. The vehicle does not have moving surfaces: control is provided by the displacement of the battery package that varies the position of the centre of mass. The wings aerofoil is based on the Eppler E838 Hydrofoil. The aerofoil has the maximum thickness 18.4% at 37.2% chord and maximum camber 0% at 46.5% chord. The arrangement of the internal sectors is visible in Figure 2a,b. The buoyancy system is contained in the buoyancy control bay: it accommodates the buoyancy motor and the oil tank and provides longitudinal balance to the system by adjusting the level in the reservoir. The bladder is contained in the hydrodynamic fairing, in contact with the open water. The fairing is not a critical structural part—it has the task of not disturbing the hydrodynamic flow of the fuselage [28].

#### 1.1.4. Conventions

We introduce, for clarity, the mathematical conventions and symbols that will be used in the subsequent discussion (see Figure 3a,b):

where: *α* (or *φ*) is the angle between the *x* axis and the *N* axis.*β* (or *θ*) is the angle between the *z* axis and the *Z* axis.*γ* (or *ψ*) is the angle between the *N* axis and the *X* axis.

For the rotation matrix, we have:(1)$$R=\left(\begin{array}{ccc} 0 & -\zeta_{z} & \zeta_{y}\\ \zeta_{z} & 0 & -\zeta_{x}\\ -\zeta_{y} & \zeta_{x} & 0 \end{array}\right)$$
where ζ is the parameter vector [29].

## 2. Materials and Methods

### 2.1. The Buoyancy System

#### 2.1.1. Basic Concepts

Gliders are controlled through hydrostatics (vertical forces) and manipulate hydrostatic balances in order to accomplish roll and pitch of the vehicle. Stability of the vehicle is a major critical factor: a stable vehicle has the centre of gravity below the centre of buoyancy. In this configuration, the weight of the vehicle creates a restoring moment to add stability to the vehicle. Roll and pitch on the glider is accomplished by moving the battery pack. Figure 4 below displays a basic concept of a buoyancy system for the glider [30,31,32,33,34,35,36]. 

The system is extremely simple: while descending, hydraulic fluid moves from the external inflatable bladder, which produces a high pressure in the internal reservoir, which is at a low pressure through a valve: the decrease in volume of the bladder creates an increase in density, causing negative buoyancy [37,38,39,40,41,42,43,44]. 

While ascending, hydraulic fluid moves from internal accumulator to the external inflatable bladder through the pump. The increase in volume creates a decrease in density causing positive buoyancy. The seawater also flushes out the open hydrodynamic fairing of the vehicle, aiding it to rise to the surface. For neutral buoyancy, the vehicle must have a density equal to seawater [45,46,47,48,49,50,51,52]. 

#### 2.1.2. The System Prototype

Our group has developed a technology demonstrator (see Figure 4) of the buoyancy system to validate the related technology and then to test it. To reduce the force required to actuate the oil piston, which pushes the oil in the bladder at high pressure, is necessary to reduce the piston surface (diameter) and increase the stroke: so, the buoyancy engine resembles a “shotgun”. An open-loop stepper motor was used to drive the screw inside the actuator that, in turn, pushes the piston. Two solenoid valves regulate the flow of oil into the bladder [53,54,55,56,57]. 

The first problem was the occurrence of actuator buckling: under the push of the engine, the probability of a part bending is high, thus deforming the thread and jamming the mechanism. The problem was solved by constructing a rigid cage with four struts that support the piston’s push load, leaving the screw only with the rolling friction load. In the early project development stages, the workgroup was oriented to use a centrifugal pump for all drives: this technology however did not allow us to create strong pressure differences; the need to use a more powerful engine was also highlighted because the prevalence was too low: this would have led to an excessive battery consumption. The second solution was to use volumetric pumps in order to obtain greater differences in pressure (even ones considerably higher than needed). Unfortunately, these would require too much power and are too heavy for our small vehicle [58,59,60,61,62]. 

At this stage of development, we have also thought to use the oil tank only as a passive fluid reservoir and trimmable counterweight of the payload and as an active actuator for longitudinal stability. The long travel of the piston (forward or backward) ensures the necessary variation of the bladder volume for manoeuvrability [63]. 

Now, the system prototype seems to work correctly and promises new developments: it is under in several fatigue cycle trials in order to investigate the parts more prone to failure as a sort of burn-in test. Our prototype (breadboard) is presented in Figure 5.

### 2.2. Buoyancy

Archimedes’ principle is the main concept underlying the buoyancy of underwater vehicles. When a vehicle is submerged in water, a buoyant force acts on the body vertically upward due to the pressure forces below the submerged body being greater than the pressure forces above. The buoyant force results in a value equal to the weight it displaces [64].

#### 2.2.1. Static Buoyancy

Now, consider our drone (see Figure 6): it is in steady state.

In these conditions, the total weight *W_TOT_* of the glider is given by:(2)$$W_{TOT}=-W_{DW}+\;B_{GB}+B_{BB}$$
where:

$W_{TOT}$ = Net total “weight” in the water.

$W_{DW}$ = Dry Weight of the glider. 

$B_{BB}$ = Buoyancy of the oil bladder.

$B_{GB}$ = Buoyancy of the naked glider.

The expression of dry weight is:(3)$$W_{DW}=W_{BA}+W_{OT}+W_{GB}$$
where:

$W_{BA}$ = Weight of the battery pack. 

$W_{OT}$ = Weight of the oil tank.

$W_{GB}$ = Weight of the naked glider (without oil tank and batteries). For “dry weight”, we mean the weight of the vehicle out of the sea, without the hydrostatic force.

So Equation (2) becomes:(4)$$\sum F_{z}=\;W_{BA}+W_{OT}+W_{GB}+B_{GB}+B_{BB}=0$$

This series of equations will be useful later to establish the drone descent attitude.

#### 2.2.2. Dynamic Balance on the Vertical Plane

Here, the drone dive (or emersion) is examined at constant speed: it is in steady-state gliding, the geometry of total forces is explained in Figure 7a. Figure 7b shows the Eppler E838 characteristic “*Cl vs. Alpha (=angle of attack*
$\alpha_{att}$)”.

At equilibrium, for the dynamics on the vertical plane at constant speed we have:(5)$$\vec{W_{TOT}}+\vec{L}+\vec{D}=0$$

The expression for the lift is:(6)$$L=\frac{1}{2}\rho v^{2}SC_{L}$$

According to the Eppler E838 characteristic “*Cl vs. Alpha*” (Figure 7b) when the angle of attack $\alpha_{att}$ = 0° (is null) the $C_{L}$ is zero so that the lift force L is null. This shows that the drone cannot progress horizontally at constant speed (straight and level): the only mission profile allowed is a sawtooth curve [65]. 

#### 2.2.3. Glider Trajectory

Because its motion is due to the difference between the forces of weight and buoyancy, the glider is unable to proceed straight and level, thus being forced to follow a dive/climb trajectory made smooth by the wings. Moreover, unlike gliders in air, AUVs can have ascending glide slopes if the net buoyancy is positive, producing a negative sink rate. 

The buoyancy engine of the glider allows changing its net buoyancy into alternating positive and negative states, thereby imparting it with the ability to string together a succession of descending and ascending glide slopes referred to as a sawtooth glide.

The behaviour of the vehicle is considered in case of a simple glide slope (refer to Figure 7a):(7)$$\begin{array}{c} Lift=L=qSC_{L}\\ Drag=D=qSC_{D}\\ Pitching\;moment=𝕞=qScC_{m} \end{array}$$

where:

$q=\frac{1}{2}\rho v^{2}$: is the dynamic pressure.

S: is the characteristic area.

c: is the mean aerodynamic chord.

$\alpha_{att}$: is the angle of attack.

For the other coefficients, we have:(8)$$\begin{array}{c} C_{L}\left(\alpha \right)=C_{L}^{\alpha }\cdot \alpha_{att}\\ C_{D}\left(\alpha \right)=C_{D}^{0}+C_{D}^{\alpha }\cdot \alpha_{att}^{2}\\ C_{m}\left(\alpha \right)=C_{m}^{\alpha }\cdot \alpha_{att} \end{array}$$

The coefficient of drag $C_{D}$ is composed of two members: the first $C_{D}^{0}$ is insensitive to the angle of attack and is constant; the second one ($C_{D}^{\alpha }\cdot \alpha_{att}^{2}$) is instead a function of the square of the angle. Note that the zero lift coefficient $C_{L}^{0}=0$ because the wing profile that has been chosen for our project is symmetrical (type *Eppler 883*). Now is necessary to separate the contributions of the fuselage (body) and of the wings, for the three factors of lift, friction and pitching moment; so the expression is:(9)$$\{\begin{array}{l} L=\;L^{b}+L^{w}\\ D=\;D^{b}+D^{w}\\ 𝕞=\;m^{b}+m^{w} \end{array}$$

According to the Navier-Stokes (approximated) equations and the simplifications above cited, the previous system of equations becomes:(10)$$\{\begin{array}{l} L=q\sqrt{V^{3}}\left\{C_{L}^{b_{\alpha }}\cdot \alpha_{att}+\frac{S_{w}}{\sqrt{V^{3}}}\cdot C_{L}^{w_{\alpha }}\cdot \alpha_{att}\right\}\\ D=q\sqrt{V^{3}}\left\{\left[C_{D}^{b_{0}}+C_{D}^{b_{\alpha }}\cdot \alpha_{att}^{2}\right]+\frac{S_{w}}{\sqrt{V^{3}}}\left[C_{D}^{w_{0}}+C_{D}^{w_{\alpha }}\cdot \alpha_{att}^{2}\right]\right\}\\ 𝕞=q\sqrt{V^{3}}\left\{C_{m}^{b_{\alpha }}\cdot \alpha_{att}-c_{m}\frac{l_{cb/ac_{w}}\cdot S_{w}}{c\cdot \sqrt{V^{3}}}\cdot C_{L}^{w_{\alpha }}\cdot \alpha_{att}\right\} \end{array}$$
where:

$S_{w}$ is the wing area.

$l_{cb/ac_{w}}$ is the distance between the mean aerodynamic and the center of buoyancy.

$c_{m}$ is a non-dimensional coefficient.

This parameter is necessary to know the exact attitude and therefore the $\alpha_{att}$ to obtain a constant descent profile [66].

#### 2.2.4. Gliding Forces

In order to allow the vehicle to glide, it is necessary to create a differential buoyancy force and therefore the balance is not null: from the buoyant force expression, the change in volume needed for the buoyancy engine for a full dive is calculated as:(11)$$\sum^{\;}F_{Z}{}_{umbalanced}=\Delta B_{BB}\neq 0$$

In which $\Delta B_{BB}\neq 0$ is the buoyancy force due to the difference of volume of the bladder:(12)$$\sum^{\;}F_{Z}{}_{umbalanced}=\Delta B_{BB}=\frac{1}{2}\;\rho g\cdot \Delta V_{bladder}$$
where:

ρ = Seawater density (average 1.025 kg/l).

g= Gravity approx. to 9.81 m/s^2^.

$\Delta V_{bladder}=$ Volume difference of the bladder.

The relationship between difference volume of the bladder and buoyancy force is:(13)$$\Delta V_{bladder}=\;\frac{2}{\rho g}\;\cdot \Delta B_{BB}$$

#### 2.2.5. Restoring Moment on the Vertical Plan

The position of the center of mass is given by:(14)$$r_{CG}=\frac{\int r\cdot \rho \left(r\right)dV}{\int \rho \left(r\right)dV}$$
where:

ρ(r): is the density.

dV: is the considered volume.

*r*: is the distance considered from the reference frame.

The gliders masses define above, is:(15)$$r_{CG}=\frac{\sum_{i}r_{i}\cdot W_{i}}{\sum_{i}W_{i}}=\frac{r_{BA}\cdot W_{BA}+r_{OT}\cdot W_{OT}+r_{GB}\cdot W_{GB}}{W_{BA}+W_{OT}+W_{GB}}$$

The vectorial balance of the forces becomes:(16)$$\{\begin{array}{l} F_{gravitational}=W_{DW}\left(R^{T}z\right)\\ F_{buoyancy}=-V_{Dispalacement}\cdot g\left(R^{T}z\right) \end{array}$$
where

$V_{Dispalacement}$ is the volume of the drone (submerged).

$R^{T}$ is the rotation function around the reference frame.

z is the upward direction.

When the geometrical centre of the body is offset from the CG frame, the resulting torque $T_{G}$ is given by:(17)$$T_{G}=r_{CG}\;\times \;W_{DW}\left(R^{T}z\right)$$

The expression of $r_{CG}$ is:(18)$$r_{CG}=\;\left(\begin{array}{c} x_{CG}\\ y_{CG}\\ z_{CG} \end{array}\right)$$

The expression of ${\hat{r}}_{CG}$ is:(19)$${\hat{r}}_{CG}=\;\left(\begin{array}{ccc} 0 & -z_{CG} & y_{CG}\\ z_{CG} & 0 & -x_{CG}\\ -y_{CG} & x_{CG} & 0 \end{array}\right)\;$$

Therefore, we have:(20)$$T_{G}=W_{DW}\cdot {\hat{r}}_{CG}\left(R^{T}z\right)$$

According to which, when the body frame is coincident with the centre of the figure:(21)$$W_{DW}\cdot r_{CG}=r_{GB}\cdot B_{GB}+r_{BB}\cdot B_{BB}$$

Moreover, the resulting balancing torque is
(22)$$T_{G}=\left(r_{GB}\cdot B_{GB}+r_{BB}\cdot B_{BB}\right)\cdot g\left(R^{T}z\right)$$

This parameter is necessary to know the exact torque and therefore the position to which the servomechanism to move the battery pack will have to obtain a constant descent profile [67].

#### 2.2.6. Uncertainty in the Depth Control

In this part, the problem of uncertainty in controlling the depth of the drone is examined. In the case where the drone has a very low vertical speed, due to the high viscosity of the water, the vehicle stops (and therefore stabilizes itself in the vertical plan) in a few centimeters of water, so this is not a problem.

Consider the case in which the drone is gliding, at a constant speed ($V_{z}<0$): at a depth of +5 m compared to the “target depth”, the bladder swells up to make the drone assume a neutral buoyancy. From a dynamic point of view, our simulations show us that the drone will behave like a mass-spring-damper system, whose temporal behavior is described in Equation (23) and visible in Figure 8 (blue line).
(23)$$\{\begin{array}{c} z_{CG}\left(t\right)=Ze^{-\zeta \omega_{n}t}\cdot \cos\left(\sqrt{1-\zeta^{2}}\omega_{n}t-\Phi \right)\\ \omega_{n}=2\pi f_{n} \end{array}$$

The maximum precision in the measurement of depth is not given by the precision of the instrument, in this case a piezoelectric depth meter can be accurate up to 2.5 cm, but in the architecture of the drone. Because of all the possible attitudes of the vehicle, it is not known if the transducer is placed higher or lower than the centre of gravity: so, unless more than one transducer is installed (which is impractical) the only consideration is that the maximum distance at which the transducer should be placed from the centre of gravity is precisely 49.7 cm, so, the maximum range of precision obtainable from the instrument is:(24)$$\delta_{z}=0.497\;m$$

As shown in Figure 8 (blue line) the natural dynamic stability of the vehicle, i.e., the oscillation within the error band, is obtained after a time $T_{normal}=9.78\;s$ which is not acceptable (see Equation (25)). It is not acceptable for two reasons: first of all because of the long damping time of the oscillation; secondly for the overshoot of the “target depth”: the depth limit may have been placed not only by the type of mission but also by the nature (orography) of the seabed. Therefore, such behavior could lead to a collision of the vehicle with the seabed itself or a known obstacle, thus leading to damage:(25)$$\{\begin{array}{c} z_{CG}{\left(t\right)}_{normal}\leq \delta_{z}\\ \;T_{normal}=9.78\;s \end{array}$$

The solution is to use the bladder as a “hydrostatic parachute” that changes the damping conditions of the system. In this case the simulation have evidenced that, if you place the bladder at maximum buoyancy (Figure 8—red line) shortly after reaching “+5” depth, the resulting behavior of the vehicle is visible in Figure 8 (azure line). 

In this case the tolerance band is reached in $T_{compensated}=4.85$ s (see Equation (26)), half the previous time and there is no danger of overshooting the depth, thus keeping us always in safe conditions:(26)$$\{\begin{array}{c} z_{CG}{\left(t\right)}_{compensated}\leq \delta_{z}\\ T_{compensated}=4.85\;s \end{array}$$

From a mathematical point of view, the system changes the damping factor, increasing it considerably in fact, $\;\zeta_{normal}<\zeta_{compensated}$. The model parameters are shown in Table 1 below.

#### 2.2.7. Dynamic Simulation

In the section, we simulate the behavior of the Eppler 838 aerofoil and the fuselage to evaluate, before a detailed and expensive but more sophisticated 3D simulation, if the proportions and dimensions of the drone fall within the range of measures evaluated as a requirement. For this purpose, the program “JavaFoil—Analysis of Airfoils” [68] is used. JavaFoil is a program which uses several traditional methods for aerofoil analysis. The backbone of the program consists of two methods:The evaluation of the potential flow. The analysis is done with a higher order panel method (linear varying vorticity distribution) and it calculates the local, inviscid flow velocity along the surface of the aerofoil for any desired angle of attack.The evaluation of the boundary layer. The analysis is steps along the upper and the lower surfaces of the aerofoil, starting at the stagnation point, solves a set of differential equations to find the various boundary layer parameters, according to the integral method.

The equations and criteria for transition and separation are based on the procedures described by Eppler. A standard compressibility correction according to Karman and Tsien has been implemented to take moderate Mach number effects into account. Usually this means Mach numbers between zero and 0.5 [69,70,71,72].

The simulation shows the critical parameters for the hydrodynamic behaviour of the model: the results are given in Table 2 below [73,74,75,76,77,78].

The trend of the flow lines with $\alpha_{att}=3.3$° around the body and the wing are visible in Figure 9a,b, below.

For completeness, in order to be able to express the induced resistance and the stall delay, the behavior of the aerofoil at different angles of attack is provided in Figure 10.

## 3. Conclusions

This paper reports part of several preliminary studies of the Underwater Drones Group of the Università degli Studi “Roma Tre” Science Department and follows the route traced on several conference papers presented at the IEEE International Workshop on Metrology for the Sea (MetroSea); this part is dedicated to the design and engineering study of the part relating to the UAV buoyancy.

This paper highlights the large series of considerations and structural dimensions, going down in great detail, of the Underwater Glider Mk. III that is currently in an advanced development phase. The real novelties of this work are highlighted in the development due to two strong constraints that our group inserted during the design of this AUV: the first is to evaluate the project always under the most conservative (pessimistic) operating conditions; the second is to evaluate how any changes made to the subsystem in development (in our case the buoyancy s/s) is reflected (and forced) on all the other parts (or subsystems). All the results both from the partial simulations and from the construction and testing of subsystems will then be used in the operating vehicle. 

The first section is dedicated to the design and engineering study of the part relating to the UAV buoyancy. In the first section, the architecture, the internal arrangement of the sub glider, the type of mission profile and the maximum requirements for the performances are broadly described. In the second part, the buoyancy system is described from an engineering-construction point of view: the solutions developed and implemented in a working prototype were illustrated.

The last part describes the mathematical requirements for sizing the vehicle. Firstly, the static requirements that are used to determine the mechanical and dimensional sizing of the buoyancy engine are examined. Then the dynamic stability (on the vertical plane) of the vehicle is analysed: this quantifies the forces involved during the “glide”. The trajectory is analysed to decide the attitude and the angle of attack: the latter is necessary, in stationary conditions, to determine the work point of the profile or the position of the profile in the diagram “*Cl vs. Alpha*”. In order to have a constant attitude, it is necessary to balance the moments in the vertical plane so that, once the wing profile is “started” and in progressive acceleration (i.e., while is nearly to reach the terminal velocity) the pitch up effects must be compensated by the movement of the centre of gravity. An evaluation of the uncertainty in the depth control is also provided.

Lastly, a simplified simulation is introduced in order to observe the hydrodynamic behaviour of the fuselage (limited to the profile) and of the aerofoil at different angles of attack, to highlight the stall characteristics.

## Figures and Tables

**Figure 1 sensors-19-01831-f001:**
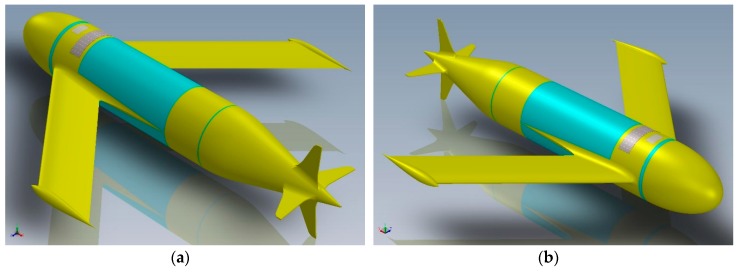
Perspective view of the Underwater Glider Mk. III. (**a**) Rear/port; (**b**) front/starboard.

**Figure 2 sensors-19-01831-f002:**
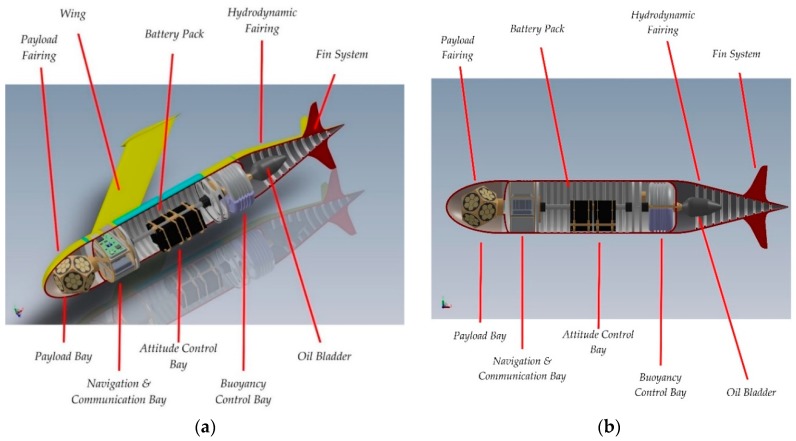
Underwater Glider Mk. III cutaway: (**a**) fuselage prospective section; (**b**) fuselage sagittal section.

**Figure 3 sensors-19-01831-f003:**
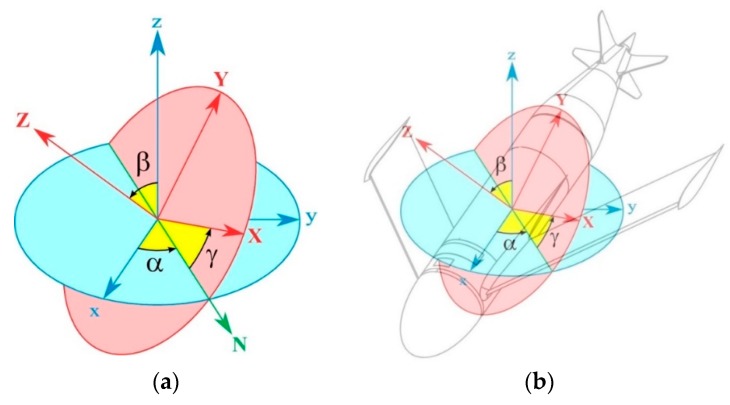
The Euler angles. (**a**) Body frame (blue) and reference frame (red); (**b**) The body frame referred to the drone.

**Figure 4 sensors-19-01831-f004:**
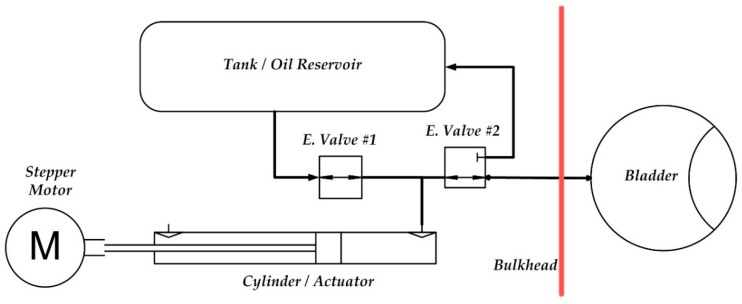
Basic scheme of the buoyancy system.

**Figure 5 sensors-19-01831-f005:**
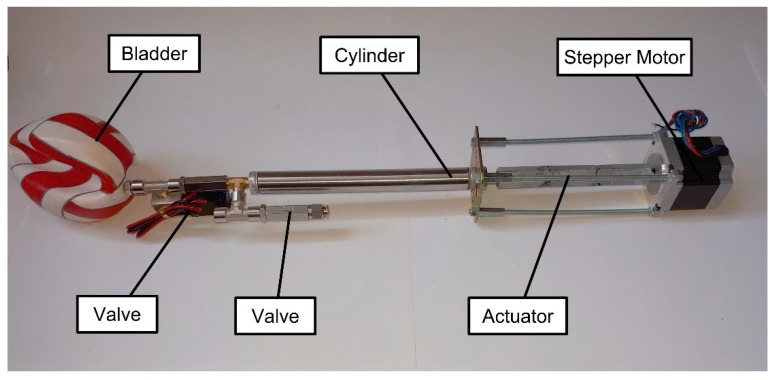
Buoyancy engine prototype.

**Figure 6 sensors-19-01831-f006:**
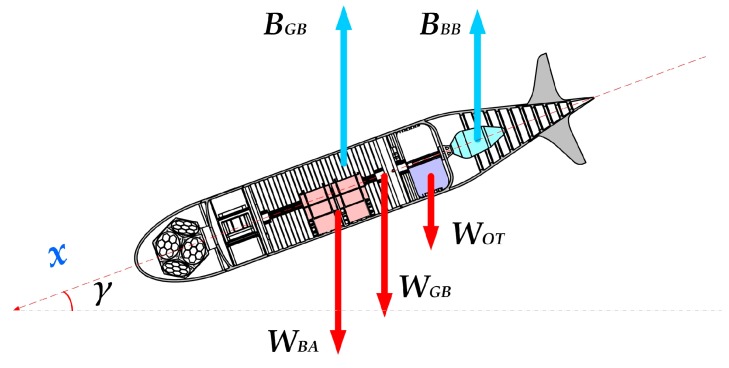
Drone in buoyancy Balance.

**Figure 7 sensors-19-01831-f007:**
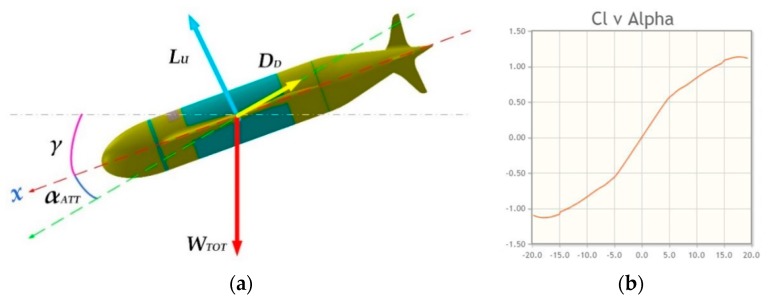
Dynamic balance of the forces (**a**) Drone geometrical balance of the forces; (**b**) Cl vs. Alpha (=angle of attack, $\alpha_{att}$ diagram for Eppler E838 aerofoil.

**Figure 8 sensors-19-01831-f008:**
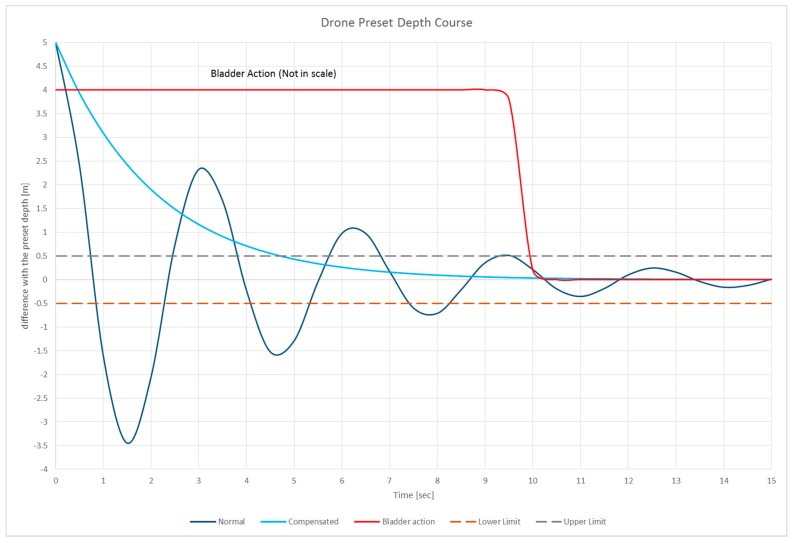
Drone depth dynamic behavior: in the blue line the normal (uncompensated) damping; the azure line the compensated behavior; dotted lines are the upper and lower limit. The red line (out of scale) is the bladder’s “hydrostatic chute” action.

**Figure 9 sensors-19-01831-f009:**
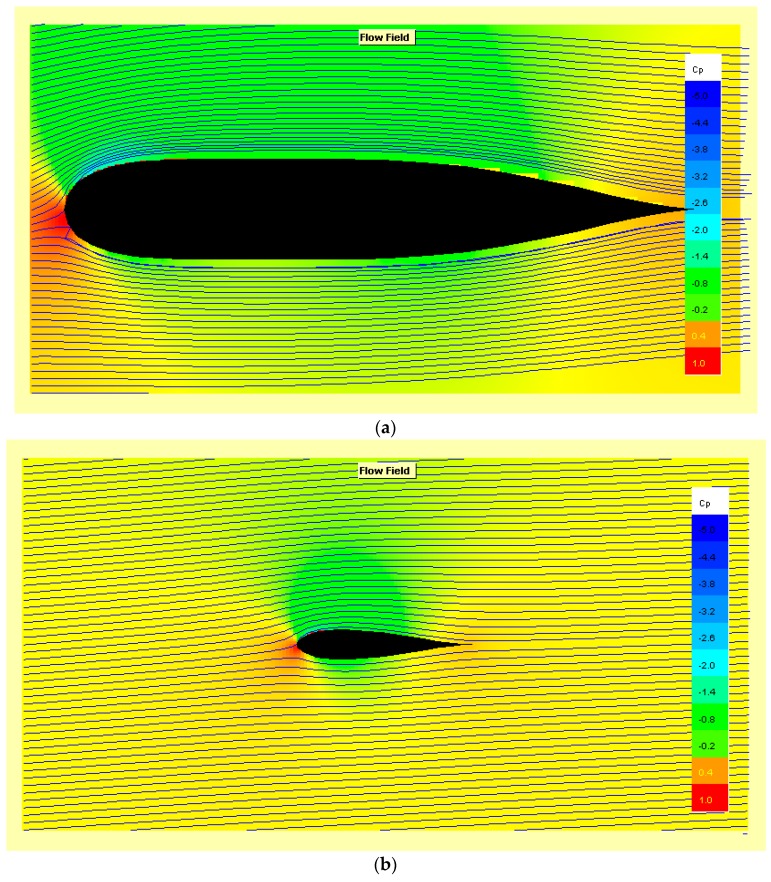
Flow field and pressure gradient of conditions: Speed = 0.52 kts, R_e_ = 10^4^ and $\alpha_{att}=3.3{}^{\circ}$, (**a**) Body profile of the drone; (**b**) Eppler 838 Aerofoil.

**Figure 10 sensors-19-01831-f010:**
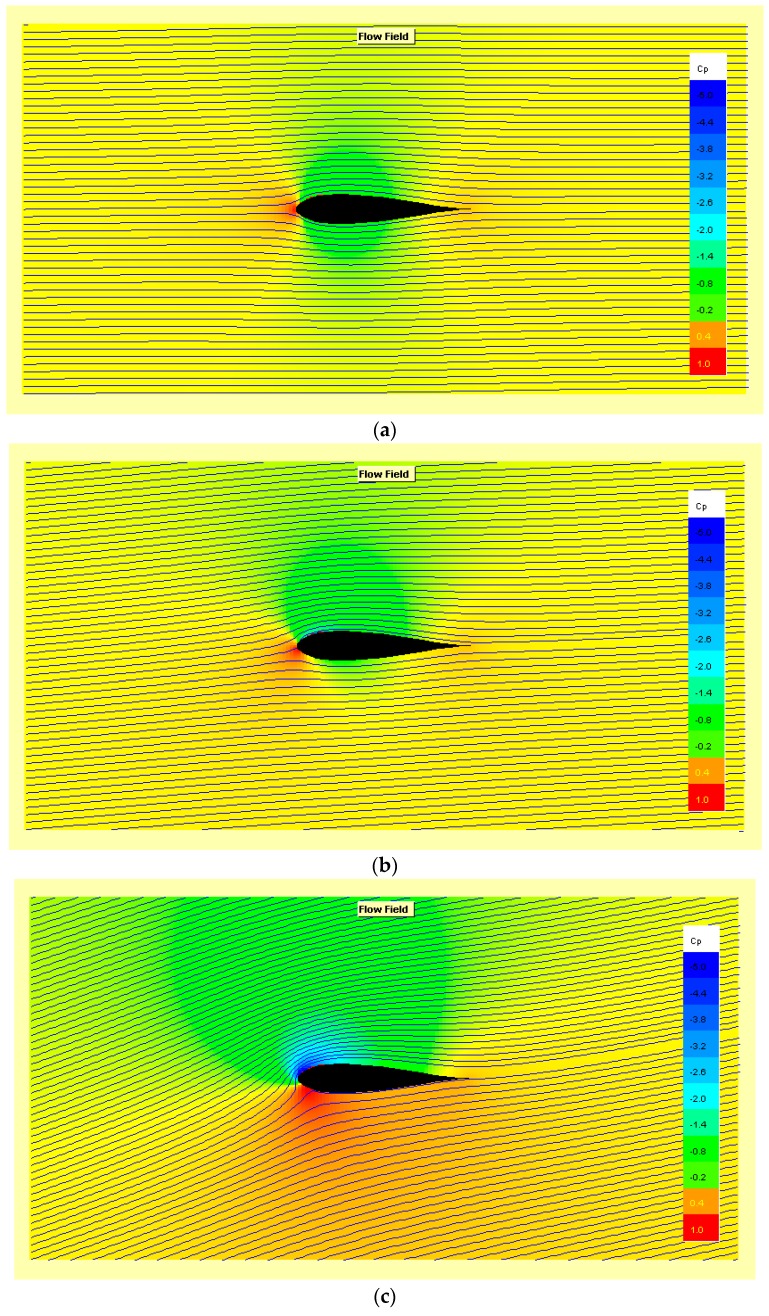
Flow field and pressure gradient of conditions (Speed = 0.52 kts, R_e_ = 10^4^) are shown at the following $\alpha_{att}$: (**a**) 0.5° (**b**) 4.0° (**c**) 15.0°.

**Table 1 sensors-19-01831-t001:** Model parameters.

$f_{n}$	$\zeta_{normal}$	$\zeta_{compensated}$	Z
1	0.0794	0.15779	4.972

**Table 2 sensors-19-01831-t002:** Body and wing critical parameters [Speed = 0.52 kts, R_e_ = 10^4^ and $\alpha_{att}\;=\;3.3{}^{\circ}$].

$C_{L}^{b_{\alpha }}$	$C_{L}^{w_{\alpha }}$	$C_{D}^{b_{0}}$	$C_{D}^{w_{0}}$	$C_{D}^{b_{\alpha }}$	$C_{D}^{w_{\alpha }}$	$C_{m}^{b_{\alpha }}$	$c_{m}$
0.3997	0.4085	0.01746	0.01676	0.00018	0.001515	0.005	0.4752

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
