# Peer review of "High Accuracy Buoyancy for Underwater Gliders: The Uncertainty in the Depth Control†"

_sensors, 2019, doi:10.3390/s19081831_

Round 1
Reviewer 1 Report
Dear Authors,
The paper deals with an argument not common in literature and extremely articulated under many profiles: the design and development of a special Autonomous Underwater Vehicles AUV for underwater applications is extremely complex and the idea of the glider poses further difficulties with respect to more usual solutions.
The paper tries to solve some of these problems suggesting many solutions starting from the mechanical aspects up to electronic and software ones presenting the project in a holistic way. In the paper are developed the main requirements and the boundary conditions in order to design the buoyancy system and to develop the mathematical conditions that define the main parameters.
The paper is very pertinent to the Special Issue and shows a very great work done well signalled from the dimension of the research and development project, and this is surely a point of honour for your team and demonstrates a great experience in this research field. A so active and complex combination of hardware and software solutions joined to experimental drone active in a very aggressive maritime environment is rare to see. Moreover, under an engineering point of view, the strong reference to the regulations is impressive and shows how much this work is strongly oriented to real realization even thinking to the further effort, made from the Authors, to develop a methodology for analysing the uncertainty in the depth control for underwater gliders.
In my opinion, the paper is well written definitely deserves to be published but, in order to further improve it, it is necessary to tackle some minor aspects:
1. Ref. line 62 - In “The Mk. III architecture”, please change the sentence “The Mk. III sub-glider has a cylindrical …” too long and hard to read.
2. Ref. line 67 - “Eppler E838 Hydrofoil” please clarify the modalities of the choice of that particular profile
3. Ref. line 116 - Please change the sentence “At first stages of the project…” too long and hard to read.
4. Ref. line 148 - Please clarify the “Weight of the naked glider” concept: is not clear
5. Ref. line 268 - Please change the sentence “In the section, we simulate the behaviour …” too long and hard to read.
6. Improve the reference section with other suitable works about the matter
7. Correct typos and English grammatical errors.
Author Response
Dear Editor, Dear Reviewers,
First of all, I would like to express my personal thanks to the Reviewers who helped us to better focus on the innovative parts of our work. Following the changes requested by the Reviewers, we have modified the paper “High accuracy buoyancy for underwater gliders: the uncertainty in the depth control” according to the proposed suggestions. We remind you that this manuscript is an extension version of the conference paper: “A High Accuracy Buoyancy System Control for an Underwater Glider” presented in the 2nd IEEE International Workshop on Metrology for the Sea (MetroSea), Bari, Italy, 8–10 October 2018.
We answer separately to the reviewers:
Reviewer #1
We would like to thank the Reviewer #1 for having identified the spirit of our paper; anyway, in order to meet your requirements and in order to improve the readability of the paper, we have simplified and clarified several sentences in the introduction chapter and in the following chapters. Finally, we have increased the reference section by inserting other interesting works that have positively influenced our paper.
· Point: 2 - We thank the Reviewer #1 for the question that allows us to the choice of that particular profile for our vehicle.
· Point: 4 - By "dry weight" we mean the weight of the vehicle out of the sea, without the hydrostatic force. We have further explained the topic in the manuscript.
Reviewer #2
We would like to thank the Reviewer #2 for his particularly welcome comments: it allows us to better explain the innovative parts of our work. Before proceeding to the discussion of the various critical points that have been pointed out, we would like to insert a necessary clarification which, unfortunately, is weakened at the first draft of the manuscript. Nowadays there are only 3 operating sub gliders worldwide, known as: Seaglider, Spray and SLOCUM (please refer to: “Bachmayer Leonard Graver - Underwater Gliders Recent Developments and Future Applications”); all three have been developed in the USA and all three are optimized for oceanic use, unfortunately they are difficult to use in a European scenario such as the Mediterranean Sea or the proximal oceanic shores. The Underwater Glider Mk. III is the first vehicle specifically designed for these scenarios, which also allows a modular and fully customizable payload. At the time of writing, nothing similar (at a professional level) has yet been developed in Europe; moreover, the documentation relating to the 3 existing gliders (for obvious reasons) is very schematic and incomplete, and, in addition, it is partially covered by patents. For these reasons our Underwater Drones Group of the Science Department is proceeding independently in the develop of a sub glider: given the strong innovation component of the vehicle, it is not possible to develop a complete dynamic model as the project is refined with time by means of "successive approximations". Every structure and subsystem still undergoes extensive modifications as it is closely related to the others: so the project cannot be thought as composed by separate modules, subsystems or watertight compartments. Nevertheless, we have made a large series of considerations and structural dimensions, going down in great detail, but, to do so, it is necessary to place two strong constraints: the first is to evaluate the project always in the most conservative (pessimistic) operating conditions; the second is to evaluate that any changes are made to the subsystem we are developing (in our case the buoyancy s/s) is reflected (and forced) on all the others parts or subsystems. This is why our paper shows how we are refining the project and how much we are able to control the large number of parameters. All the results both from the partial simulations and from the construction and testing of subsystems that will then be installed in the operating vehicle. The enormous complexity of the study therefore prevents the possibility of creating a mathematical model of the drone since, undoubtedly, our work group is following paths that have never been traveled before. Despite these enormous difficulties, we have succeeded in refining the development of the vehicle very well, so we want to offer the results of all the problems of buoyancy control in this paper.
Therefore, we have integrated and fulfilled all the specific requests of the Reviewer #2. In particular:
a) Point 2.1.2 – the presented prototype is still under trial in a hyperbaric chamber: the paragraph just wants to illustrate with what means we wanted to solve the problem of the buoyancy subsystem. The chapter also specifies the type of system chosen or the closed system: generally submarine boats use an open system (technically much simpler) for buoyancy regulation, which works by opening some tanks to sea water. In our case this is inapplicable as the danger due to the contamination of the salt is always present and would require excessive maintenance.
b) Point 2.2.7 – As said at the beginning, the modeling of the entire vehicle is extremely complex: in this way, to simplify it, we wanted to study the behavior of the wings separately and look for the point of instability in which we have the detachment of the laminar flux layer: this to establish the limit angle of attack. It will then be fixed as limit of the navigation system. Also in this case, we placed ourselves in an extremely cautious condition in order to calculate the forces acting on the vehicle. Please note that, in the real case, the presence of the tail will have a beneficial effect on hydrodynamics and stability: to fully examine its contribution, it will be necessary thereafter to develop a more sophisticated model that will also take into account the "washdown effect" of the wing. For the moment, we have limited ourselves to finding the worst dynamic conditions. The same argument is applicable to the wing: as we can see, we have not deliberately considered the "Kalman’s fitting" (the extended wing-fuselage connection clearly visible in fig 1 and 2) which always has a positive effect on hydrodynamics: this operation was also done for the purpose of putting us here in a conservative condition. In the future, when the project is more defined, we will bring to your attention a more detailed model.
c) See point b)
d) At the moment it is impossible to even think about developing a simulation as the project, in its various parts is not yet frozen, as specified in my opening discussion.
e) We agree with Reviewer #2: we have accepted the suggestion not to use the first person: in light of this, we have completely revised the paper, making it more correct and smooth in reading.
f) We agree with Reviewer #2: the conclusions were suitably modified to highlight the innovative part of the work and new clarifications within a larger architecture of many explanations.
(To all Reviewers)
Finally, we have also improved the quality and readability of all the figures in the paper, as well as improving the English style of the text and carefully eliminating every misprint and have integrated the bibliography section and modified the style according to the standard format.
My best Regards
Fabio Leccese
Reviewer 2 Report
Congratulations for your paper: Let me address some some doubts and recommendations:
- point 2.1.2. A real prototype is presented, but there are no measurements and evaluations of the system. Just a description. There is no comparison with the following simulations of the control system.
- point 2.2.7: dynamic simulations does not consider the full AVU body, just the main part of the fuselage. Why the wings and tail are not included?
The relation between such simulations and the control system is not clear.
- I'm not an expert on the mathematical model of the instrument and the control system: but will be recommendable a link and comparison between simulations and real measurements on the buoyancy control system prototype.
- the writing uses the 1st person: "we use", "we simulate", etc... I would recommend a redaction using " xxx has be used", " the xxxx has been simulated", etc...
- there are no real conclusions. Conclusions just enumerate the content of the article. Please, give real conclusions about the contributions and results of the simulations.
Author Response

(The authors gave the same response as above.)
